# Exome Sequencing in Adults with Unexplained Liver Disease: Diagnostic Yield and Clinical Impact

**DOI:** 10.3390/diagnostics15162010

**Published:** 2025-08-11

**Authors:** Kenan Moral, Gülsüm Kayhan, Tarik Duzenli, Sinan Sari, Mehmet Cindoruk, Nergiz Ekmen

**Affiliations:** 1Division of Gastroenterology and Hepatology, Department of Internal Medicine, Gazi University, Ankara 06560, Turkey; dr.k.moral@gmail.com (K.M.); drmehmetcindoruk@gmail.com (M.C.); 2Department of Medical Genetics, Gazi University, Ankara 06560, Turkey; drkayhangulsum@gmail.com (G.K.); tarikduzenli@outlook.com (T.D.); 3Division of Pediatric Gastroenterology and Hepatology, Department of Pediatrics, Gazi University, Ankara 06560, Turkey; drsinansari@gmail.com; 4Department of Gastroenterology and Hepatology, Tulane University Health Sciences Center, New Orleans, LA 8035, USA

**Keywords:** exome sequencing analysis, rare liver diseases, genomics in hepatology, undiagnosed liver diseases

## Abstract

**Background:** The etiology of liver disease remains unidentified in approximately 30% of patients, presenting a persistent diagnostic challenge. While whole-exome sequencing (WES) is well established for identifying rare genetic conditions in pediatric populations, its utility in adult hepatology is less defined. This study aimed to evaluate the diagnostic value of WES in adults with unexplained liver disorder. **Methods:** Fifty-three Turkish adult patients with idiopathic liver disease underwent a comprehensive clinical evaluation and WES at Gazi University Ankara in 2024–2025. The cohort included individuals with idiopathic cholestasis (6/53, 11%), hepatic steatosis (28/53, 53%), unexplained elevated liver enzymes (12/53, 23%), and cryptogenic cirrhosis (7/53, 13%). All patients had inconclusive results from prior standard investigations. **Results:** ES yielded a definitive molecular diagnosis in 11% (6/53) of cases. Definitive diagnoses were distributed across the following disease categories: idiopathic cholestasis (*n* = 1), hepatic steatosis (*n* = 1), elevated liver enzymes (*n* = 2), and cryptogenic cirrhosis (*n* = 2). Pathogenic variants were detected in the *ABCB4*, *AGL*, *APOB*, *CP*, and *MTTP* genes. One patient was identified with mosaic Turner syndrome. **Conclusions:** This study highlights the role of rare genetic variants in the etiology of unexplained liver disease in adults. Integrating whole-exome sequencing into hepatology practice can uncover novel disease mechanisms and improve diagnostic yield, informing more precise patient care.

## 1. Introduction

Chronic liver disease is a significant global health burden with increasing mortality rates. Despite substantial advances in the identification of liver disease etiologies in recent decades, the underlying cause remains unknown in 14–30% of patients with liver disease [1]. In recent years, the use of gene analysis in clinical practice has gained significant attention. Genomic technologies have advanced significantly, enabling the analysis of chromosomal microarrays and next-generation sequencing [2,3]. Although various clinical fields utilize genomic analysis for detecting rare variants or understanding genetic complexity in various diseases [4,5,6], the application of next-generation sequencing (NGS) analysis in hepatology remains limited [7,8]. Whole-exome sequencing (WES), a widely used NGS-based test, comprehensively sequences all protein-coding regions of the genome. WES analysis offers an advantage over targeted NGS panels that focus on specific genes by enabling the investigation of newly discovered genes associated with the phenotype [9,10].Whole-genome sequencing (WGS) is another NGS approach that sequences the entire genome. WGS has the advantage over WES in its capacity to include all non-coding regions, such as introns, untranslated regions, and intergenic areas with regulatory elements that influence gene expression. Nevertheless, since most variants linked to monogenic diseases are found within coding regions, WES remains a primary method for diagnosing these conditions [11].

WES analysis should be considered in patients with hepatic disorders presenting ambiguous manifestations in early adulthood, atypical clinical presentations, syndromic features, and multisystemic involvement. In cases where abnormalities in cholestatic liver enzymes are observed, it is advisable to conduct further testing if there is no improvement following the initial treatment. Genetic variations are likely to exert a significant influence on the phenotype in cases of early-onset disease [12]. Moreover, individuals who are offspring of consanguineous marriages or have a positive family history of hepatic disease warrant evaluation [13]. In the last few years, few studies have employed WES analysis in the field of adult hepatology. One study that assessed six lean fatty liver disease patients using WES identified two rare monogenic disease-causing variants [14]. Another case series conducted by Pelusi et al. performed a WES analysis in six adult patients with cryptogenic liver disease, identifying rare genetic variants [15]. Ronzoni et al. conducted a targeted sequence analysis of 49 adult patients with fatty liver disease and found that 22% had a Mendelian disorder and 59% showed a potential influence on the clinical phenotype in selected adults with steatotic liver disease [16].

The most extensive study, conducted by Zheng et al., uncovered definitive or presumed diagnoses in 33% of patients with unexplained liver disease [17].

Consanguineous marriages are common in Turkey, with rates ranging from 22% to 24%. This high prevalence leads to an increased occurrence of congenital abnormalities and diseases inherited through autosomal recessive patterns [18]. The aim of this study was to evaluate the diagnostic utility of WES in a cohort of adult Turkish patients with unexplained liver disease. By applying WES in a clinical hepatology setting, we sought to identify monogenic causes of liver disease that may be underdiagnosed using conventional approaches, assess the clinical relevance and phenotype correlation of detected variants, and explore the impact of WES findings on clinical management. Given the high rate of consanguinity in the Turkish population, this study also aimed to highlight the potential value of genomic diagnostics in populations with a greater prevalence of recessively inherited disorders.

## 2. Materials and Methods

The study population comprised 53 individuals with liver disease who were evaluated by three hepatologists (two adults and one pediatric) and could not receive a definitive diagnosis of their disease despite extensive evaluation. All patients included in the study underwent comprehensive clinical, laboratory, and imaging evaluations to exclude the common causes of liver disease before undergoing WES. Viral hepatitis (HBV, HCV, HAV, HEV) was excluded by negative serological testing, including HBsAg, anti-HBs, anti-HBc IgG/IgM, anti-HCV, and relevant PCR assays, when indicated. Infectious etiologies, such as CMV, EBV, and HSV, were also ruled out in selected patients with compatible clinical presentations. Drug-induced liver injury (DILI) was excluded through a detailed medication history, including over-the-counter and herbal supplements, combined with temporal correlation analysis and Roussel Uclaf causality assessment method (RUCAM) scoring when appropriate. Alcohol-related liver disease was ruled out based on a negative history of alcohol use, corroborated by family reports and clinical documentation. Autoimmune liver diseases, including autoimmune hepatitis, primary biliary cholangitis, and primary sclerosing cholangitis, were excluded based on serological tests, including ANA, SMA, anti-LKM1, anti-SLA, AMA, p-ANCA, and elevated serum IgG levels. Metabolic liver diseases, such as Wilson’s disease, hemochromatosis, and alpha-1 antitrypsin deficiency, were excluded using ceruloplasmin, 24 h urinary copper excretion, hepatic copper content at the time of biopsy, transferrin saturation, ferritin, and serum alpha-1 antitrypsin levels, and phenotyping was performed when necessary (Appendix A). Liver imaging (ultrasound, abdominal tomography (CT), or MRCP (magnetic resonance cholangiopancreatography)) and liver biopsy were performed when indicated to support the exclusion of known etiologies further. Fourteen patients—six with idiopathic cholestasis and eight with mixed-pattern liver enzyme elevation—underwent both MRCP and CT imaging, which revealed no specific abnormalities. All patients also underwent abdominal ultrasound: 28 showed evidence of fatty liver, 7 had findings consistent with cirrhosis, and the remainder had no notable abnormalities. Only patients with persistently unexplained liver abnormalities after this extensive evaluation were considered eligible for WES (Appendix A).

Patient age (age at which WES was performed), family history, history of consanguineous marriage, presence of cirrhosis, and type of hepatic presentation were recorded. Hepatic presentation was categorized as idiopathic cholestasis, hepatic steatosis, elevated liver enzyme levels, or cryptogenic cirrhosis. Family history and consanguineous marriages were considered positive based on patient reports. Cirrhosis was defined based on biopsy or liver stiffness measurements. Genetic variants were correlated with the clinical phenotypes of the patients. Genetic variants inconsistent with the clinical presentation were excluded.

### 2.1. Whole-Exome Sequencing and Data Analysis

DNA was extracted using a QIAamp DNA blood kit (Qiagen, Hilden, Germany) according to the manufacturer’s protocol, performed on a QIACube (Qiagen, Germany). For WES, the library was prepared using Comprehensive Exome Panel technology (Twist Bioscience, South San Francisco, CA, USA) and sequenced on a NovaSeq 6000 instrument (Illumina, San Diego, CA, USA). Reads were aligned to the GRCh38 Human Reference Genome using the Illumina DRAGEN Bio-IT Platform v3.9. DRAGEN pipelines were utilized to identify single-nucleotide variants (SNVs) and copy-number variants (CNVs). Ilyome software (v4.2.0) was used for variant annotation and analysis.

Variant filtering was performed using a multistep approach. First, phenotype-based filtering was applied using a predefined gene set, “abnormality of the liver; HP:0001392,” from Human Phenotype Ontology, which includes OMIM genes related to liver diseases (Appendix A). Next, frequency filtering was conducted, excluding variants with a minor allele frequency (MAF) of ≥0.01 based on gnomAD v4.1.0 and in-house frequency data. SNVs were then filtered by type, including nonsense, frameshift, splice site, stop-loss, start-loss, indels, missense, and synonymous/intronic variants predicted to affect splicing. In silico predictions were incorporated using CADD, REVEL, SpliceAI, SIFT, and PolyPhen-2 to assess the potential pathogenicity of missense and splicing variants. The remaining variants after filtering were classified according to the 2015 standards of the American College of Medical Genetics [19] (Figure 1). CNVs were filtered by population (GnomAD and DGV) and in-house frequency. After filtering, the remaining variants were classified according to the 2020 standards of the American College of Medical Genetics for CNV analysis [20]. The patients received details about how their genomic data could be utilized in this research. The genetic material was encoded alongside the patient at Gazi University Department of Gastroenterology, and access to these data were restricted to the study investigators only, as it was securely locked. The study was approved by the Gazi University Ethics Board (approval 1026) on 25 December 2023, and conducted in accordance with the Declaration of Helsinki and Istanbul. Written informed consent was obtained from all the patients.

### 2.2. Clinical Phenotypes

The following definitions were used: family history of liver disease (presence of liver disease in a first- or second-degree relative) and history of consanguineous marriage (presence of marriage between first-degree relatives or relatives within a local village).

Idiopathic cholestasis was defined as elevated levels of cholestatic enzymes, which may occur with or without increased bilirubin levels. Liver biopsy results typically reveal bile duct injury, cholestatic damage, and ductopenia at some stage, all in the absence of a specific identifiable etiology.

Hepatic steatosis was characterized by patients who either met the criteria for metabolic dysfunction-associated fatty liver disease (MASLD) along with extrahepatic manifestations indicative of a monogenic disorder or who did not align with the categories of MASLD, MASLD with increased alcohol intake, alcoholic-associated liver disease (ALD), or DILI, as outlined by the new Delphi consensus statement on fatty liver disease [21].

Elevated liver enzyme levels were characterized as either exclusively hepatic or a combination of hepatic and cholestatic elevations without any serological or pathological evidence indicative of a specific disease.

Cryptogenic cirrhosis is characterized by advanced liver fibrosis and architectural distortion for which no definitive underlying cause can be identified despite comprehensive clinical, laboratory, and imaging evaluations. Histological analysis did not reveal any specific disease aside from the presence of dense fibrous septa, small regenerative nodules, and minimal inflammation or steatosis.

Following WES analysis, a comprehensive clinical evaluation was conducted, and patients with consistent genotype–phenotype correlations, as documented in Online Mendelian Inheritance in Man (OMIM), were accepted as either definitive or possible diagnoses. All of our patients were evaluated within a multidisciplinary team consisting of pediatric and adult hepatologists, pediatric metabolic specialists, neurologists, pathologists, and clinical geneticists. The diagnostic outcomes were classified as follows: (a) a definitive genetic diagnosis was confirmed when biallelic pathogenic or likely pathogenic variants for autosomal recessive (AR) diseases, monoallelic pathogenic or likely pathogenic variants for autosomal dominant (AD) or X-linked disorders, or heterozygous mutations documented in the literature as causing liver disease, consistent with the phenotype, were identified; (b) a possible genetic contribution was considered when rare variants of uncertain significance (VUS) consistent with the phenotype and mode of inheritance were detected; (c) a negative analysis was concluded when no variants consistent with the phenotype were identified.

### 2.3. Statistics

Patients with definitive genetic diagnoses were compared with those without definitive genetic diagnosis. Categorical variables are expressed as frequencies and percentages, whereas normally distributed variables are expressed as means (±standard deviation). For non-normally distributed variables, the median (minimum–maximum) and interquartile range (25–75%) were used. The chi-squared test was used to compare the two groups. Statistical significance was set at *p* < 0.05.

## 3. Results

### 3.1. Study Cohort

A total of 53 patients were included in this study. Of these, 6/53 (11%) had idiopathic cholestasis, 28/53 (53%) presented with hepatic steatosis, 12/53 (23%) had elevated liver enzyme levels, and 7/53 (13%) were diagnosed with cryptogenic cirrhosis. The median age was 34 ± 10 years, and 18/53 (34%) participants were female. Table 1 presents the general demographic characteristics of the total patient population and the subset of patients with a definitive diagnosis. There were no statistically significant differences in family history of liver disease (*p* = 0.342) or consanguineous marriage (*p* = 0.729) between the two groups.

### 3.2. Genetic Results

WES analysis provided a definitive diagnosis in 11% of the patients. Three patients were detected with VUS that may be associated with clinical findings or pathogenic variants that are heterozygous for AR inheritance. Pathogenic variants were detected in the *ABCB4* gene in a patient with idiopathic cholestasis, in the *APOB* and *CP* genes in a patient with hepatic steatosis, in the *ABCB4* gene in a patient with elevated liver enzymes, and in the *MTTP* and *AGL* genes in a patient with cryptogenic cirrhosis. Mosaic Turner syndrome confirmed with karyotype analysis was also identified in another patient with elevated liver enzymes. Three patients were accepted as harboring a possible genetic contribution. One patient with hepatic steatosis had homozygous VUS in the *TMEM199* gene, and another with cryptogenic cirrhosis had homozygous VUS in the *FOCAD* gene. A heterozygous pathogenic variant in the *VPS33B* gene was detected in a patient with idiopathic cholestasis, which could have contributed to the clinical findings (Table 2).

### 3.3. Clinical and Genetic Findings of the Patients with VUS or Pathogenic Variants

#### 3.3.1. Idiopathic Cholestasis

Patient 1 had a history of intrahepatic cholestasis during pregnancy, cholecystitis, biliary pancreatitis, and chronically elevated cholestatic enzymes. WES analysis revealed a heterozygous known pathogenic variant of the *ABCB4* gene, c.2177C > T (p.Pro726Leu). Biallelic pathogenic variants of *ABCB4* result in progressive familial intrahepatic cholestasis 3, an autosomal recessive disorder. In contrast, individuals carrying heterozygous *ABCB4* pathogenic variants are prone to developing various conditions, including cholelithiasis, cholestasis induced by medications, pregnancy-related intrahepatic cholestasis, and cirrhosis [22]. Heterozygous carriers exhibit partially functioning proteins, resulting in milder phenotypes than compound heterozygous or homozygous carriers [23]. The findings in patient 1 were compatible with the detected variant.

Patient 2 presented with idiopathic cholestasis and underwent a liver biopsy in accordance with cholestasis. The patient’s family history was significant, with first-degree relatives having cholestatic liver disease. He had a heterozygous nonsense variant in the *VPS33B* gene, c.277C > T (p.Arg93Ter), which is considered pathogenic. This variant was described as compound heterozygous with another nonsense variant in *VPS33B* within a family affected by arthrogryposis, renal dysfunction, and cholestasis (ARCS1) [24]. While homozygous or compound heterozygous variants in the *VPS33B* gene have been reported in pediatric patients diagnosed with ARCS1 or progressive familial intrahepatic cholestasis type 12 (PFIC-12), heterozygous variants in adult patients have not been documented [25]. As previously noted, it is well established that heterozygous variants in *ABCB4* and *ABCB11* can result in milder forms or late-onset liver disease in adults [26]. We considered that the *VPS33B* heterozygous pathogenic variant may have been a contributing factor to disease development in this patient, given a positive family history, the implications of heterozygous variants in PFIC genes, and the clinical presentation of cholestasis. However, the patient may have had a second variant in the *VPS33B* gene that fell outside the limits of the test or a variant with a modifying effect.

#### 3.3.2. Hepatic Steatosis

Patient 3 was a 35-year-old man who had had elevated transaminase (alanine aminotransferase (ALT = 136 (<35 U/L)) and aspartate aminotransferase (AST = 100 (<35 U/L)) and low ceruloplasmin levels (16 mg/dl < 20 mg) since he was 13 years old. He was investigated for Wilson’s disease, but was found to be negative. Liver biopsy showed steatohepatitis with F1–F2 fibrosis. He had no anemia, but had slightly elevated ferritin levels (310 ng/mL (40–300 ng/mL) and high cholesterol levels (LDL = 190 mg/dl (<130 mg/dl) and triglyceride 270 mg/dl (<150 mg/dl)). Additionally, he experienced progressive essential tremors, dystonia, and speech difficulties, which were managed with symptomatic treatment. Furthermore, he was diagnosed with diabetes at the age of 27 (blood glucose level, 124 mg/dl and Hba1c = 6.8%). WES analysis revealed a heterozygous pathogenic variant [c.1948G > A (p.Gly650Arg)] in the *CP* gene, related to autosomal recessive aceruloplasminemia. Aceruloplasminemia is a rare genetic disorder characterized by excessive iron accumulation in the brain and other organs. It is an autosomal recessive disorder associated with a mutation in the *CP* gene that encodes the iron-oxidizing enzyme ceruloplasmin. This disorder leads to a gradual neurodegenerative process accompanied by microcytic anemia and diabetes [27]. Another key point is that *CP* variants are independently linked to hyperferritinemia, hepatic iron overload, and advanced liver fibrosis in patients with MASLD [28]. The clinical relevance of the heterozygous state is less understood: patients in this state can present with neurological manifestations or be asymptomatic, suggesting incomplete penetrance and that environmental factors play a role [29,30,31]. The patient also presented with two homozygous missense variants, VUS—p.Ala10Val and p.Gly753Glu—in the *APOB* gene, which are associated with hypobetalipoproteinemia and familial hypercholesterolemia. These variants may have been responsible for the hypercholesterolemia findings in the patient, with a possible hypomorphic effect. Additionally, there was a heterozygous VUS in the *KLF11* gene, linked to maturity-onset diabetes of the young type VII. Variants in both genes may have contributed to the observed hepatic steatosis.

Patient 4 was a lean 31-year-old patient with hepatic steatosis. He had no alcohol or drug usage. He had slightly elevated transaminases and a ceruloplasmin level of 19 mg/dl. WES analysis revealed a homozygous c.532C > T (Arg178Trp) variant in the *TMEM199* gene with uncertain clinical significance. Congenital glycosylation disorder (CGD) is a rare group of inherited metabolic diseases characterized by mild hypoceruloplasminemia. Defects occurring in cell organelles, primarily the cytosol, endoplasmic reticulum, and Golgi apparatus, result in defects in glycoprotein and glycan assembly, leading to various organ manifestations. The protein *TMEM199* plays a crucial role in the transport of glycosyltransferases within the Golgi complex [32]. Jansen J.C., et al. demonstrated in four adolescent individuals with a mild phenotype of hepatic steatosis slightly low ceruloplasmin levels and elevated aminotransferases that *TMEM199* deficiency resulting in disruption of Golgi homeostasis, consequently leading to a hepatic phenotype with abnormal glycosylation [33]. However, the clinical significance of the variant detected in the patient is unknown.

#### 3.3.3. Elevated Liver Enzymes

Patient 5 was an 18-year-old adolescent who was admitted to our department due to elevated mixed-type liver enzymes (ALT = 200 (<35 U/L), AST = 110 (<35 U/L)), alkaline phosphatase (ALP = 156 (<130 U/L)), and gamma-glutamyl transferase (GGT = 220 (<50 U/L)). Ultrasound showed normal liver parenchyma with no gallbladder stones, while the liver biopsy showed a mixed type of portal inflammation. WES analysis showed a likely pathogenic, homozygous missense variant in *ABCB4*—c.716C > T (p.Ser239Leu). This variant has been reported as homozygous in a pediatric patient with a preliminary diagnosis of PFIC [34]. It was classified as likely pathogenic and the patient was diagnosed with *ABCB4*-related PFIC3. After the start of ursodeoxycholic acid therapy, the enzymes normalized during follow-up (ALT = 30 (<35 U/L), AST = 20 (<35 U/L), ALP = 70 (<130 U/L), GGT = 90 (<50 U/L)).

Patient 6 was a 44-year-old female presenting with mixed-type elevated transaminases for 15 years. Liver biopsy revealed nonspecific portal inflammation. The patient exhibited horseshoe kidneys, polycystic ovarian syndrome, and irregular menstruation. She had infertility, but was not evaluated for this condition due to her lack of desire to bear children. Genetic analysis identified a mosaic pattern of Turner syndrome [35] (mos 45,X[23]/46,XX[17]). Turner syndrome patients are known to have endocrinological manifestations, are prone to other autoimmune diseases, and have horseshoe-shaped kidneys, ovarian manifestations, and elevated liver enzymes [36]. Hepatic manifestations include minimal abnormalities, steatohepatitis, vasculopathy, biliary involvement, cirrhosis, and nodular regenerative hyperplasia [37]. It is essential to acknowledge that mosaic forms of Turner syndrome may not exhibit the typical clinical features of this condition. Hence, in the assessment of female patients presenting with unexplained liver enzyme elevation, hepatologists must maintain a high level of awareness regarding the potential for a mosaic Turner syndrome pattern. She also had a heterozygous missense VUS (p.Leu73Val) in the *ABCB4* gene related to autosomal recessive PFIC3 and a heterozygous indel VUS (p.Asn1822_Ile1823delinsThrPhe) in the *NBAS* gene related to autosomal recessive infantile liver failure syndrome. While heterozygous carriers of *ABCB4* are prone to develop liver disease [38], the role of heterozygous *NBAS* is not well known.

#### 3.3.4. Cryptogenic Cirrhosis

Patient 7 was a 20-year-old male who presented with variceal bleeding at the age of 13. He exhibited hepatosplenomegaly, which was evaluated; however, no etiology was identified. The patient did not experience diarrhea. However, the patient manifested abdominal distension and recurrent episodes of hypoglycemia, which were investigated without revealing an endocrinological cause. He did not display intellectual disability or growth retardation. His sister and father were also diagnosed with cirrhosis, but genetic testing was not performed. A homozygous missense variant in the *FOCAD* gene, c.1507C > G (p.Pro503Ala), classified as VUS, was identified in the patient. The *FOCAD* gene has recently been associated with severe congenital liver disease, characterized by progressive hepatic dysfunction. Moreno Traspas et al. reported 14 children from 10 unrelated families with the *FOCAD* mutation. Biallelic mutations in the *FOCAD* gene have been recognized as causative agents of severe congenital liver disease, resulting in cirrhosis and extrahepatic manifestations [39]. The patient’s positive family history of cirrhosis, the early onset of cirrhosis at pediatric age, abdominal distension, and recurrent hypoglycemia suggest that this novel variant could be the cause. The patient’s slower progression compared to that seen in the literature may indicate that the identified variant has a milder effect. However, additional studies are required to clarify the pathogenicity of this variant.

Patient 8 was a 27-year-old female who presented with cirrhosis. She had low apolipoprotein B levels (<25 mg/dL, 40–100 mg/dL), LDL was 4 (50–130 mg/dL), and triglycerides were 20 mg/dL (50–150 mg/dL). She had chronic diarrhea with low vitamin A, E, and D levels. Furthermore, she had osteoporosis and blurred night vision. Genetic analysis revealed likely pathogenic compound heterozygous variants, c.59_61 + 14del and c.1946A > G (p.Asn649Ser), in the *MTTP* gene associated with abetalipoproteinemia. The c.59_61 + 14del variant disrupts the splice donor site and has not been reported before. The Asn649Ser variant has been identified in a patient with abetalipoproteinemia, and a functional study showed that it reduced the lipid transfer activity of the protein by 43%. The parental genetic analysis revealed that the variants were in a trans position. Usually, individuals diagnosed with abetalipoproteinemia have steatorrhea and vomiting. Deficiencies in fat-soluble vitamins lead to hematological (anemia, acanthocytosis, and increased risk of bleeding), neurological (neuropathy, myopathy, and spinocerebellar ataxia), and ophthalmological manifestations (retinitis pigmentosa) [40], but milder forms of abetalipoproteinemia with fatty liver disease with no neurological, hematological, or ophthalmological manifestations have been reported in the literature, most probably explained by low residual functions of the MTTP protein [41,42]. Patients with abetalipoproteinemia are prone to liver fibrosis and progression to cirrhosis [43].

Patient 9 presented with proximal muscle weakness and cirrhosis at 40 years of age. The patient exhibited no hypoglycemia or hyperuricemia; however, elevated creatine kinase levels of 1071 IU/L and hyperlipidemia were observed. He had no cardiomyopathy. Muscle biopsy findings were consistent with vacuolar myopathy, demonstrating increased glycogen content. Genetic analysis revealed a homozygous nonsense variant, c.889A > T (p.Lys297Ter), in the *AGL* gene. The patient was subsequently diagnosed with glycogen storage disease type IIIa, a condition that affects both the liver and muscles, in contrast to type IIIb, which exclusively affects the liver [44]. Glycogen storage disease type III manifests with myopathy, cardiomyopathy, and hepatic fibrosis. Affected individuals exhibit an elevated risk of developing cirrhosis, hepatocellular adenoma (HCA), and hepatocellular carcinoma (HCC) [45].

## 4. Discussion

This study demonstrated the effectiveness of WES in identifying the causes of liver disease in adults with unknown etiology. The results align with prior findings in pediatric hepatology and add to the limited data supporting the integration of WES in adult liver disease workups [46]. Of the 53 patients, 11% received a definitive diagnosis, while 3% had a possible genetic contribution. Although prior studies, such as the work by Zheng et al., reported a higher diagnostic yield (33%), differences in genetic background from different populations and patient demographics may account for this variation. Our cohort, with a mean age of 34 years, showed no statistically significant differences in diagnostic yield based on family history or consanguinity, consistent with the findings of Zheng’s study.

A separate investigation by Valenti et al. identified 15 patients with mutations in *APOB*, indicative of hypobetalipoproteinemia, in a cohort of 201 patients with MAFLD, of whom only three had a normal weight [47]. Notably, we did not detect any hypobetalipoproteinemia in our hepatic steatosis group, likely because of the absence of hypolipidemia in this subset. Additional comprehensive studies focusing on patients with lean steatosis in Turkish populations are necessary to validate this discrepancy when compared to Western-based research outcomes.

Of the nine patients with genetic findings, only six were classified as having a definitive diagnosis. The remaining three were VUS, which present interpretative challenges. While the clinical phenotypes in these cases were suggestive, the lack of segregation data, functional validation, or population-level evidence limits the confidence in causality. This highlights a broader issue in genomic medicine: VUS are common, particularly in populations that are underrepresented in global genomic databases. WES can identify rare variants; however, when data on these variants are limited, they are classified as VUS. VUS are frequently encountered during WES analysis and should be interpreted in the context of the patient’s clinical manifestations [47]. Most genotype–phenotype correlations pertain to well-established phenotypes, yet the complete spectrum of phenotypic expression for most genetic modifications remains to be thoroughly characterized. Various Mendelian traits can result in different types of hepatopathies. The variability in genotype–phenotype correlations and the potential for mild or late-onset clinical findings can be attributed to several factors, particularly in heterozygous conditions. These factors include genetic polymorphisms, epigenetic influences, and other contributing elements. The absence of functional analysis complicates the ability to draw definitive conclusions; therefore, further large-scale studies are necessary to classify these variants as either “benign” or “pathogenic.”

On the other hand, WES does not assess all genetic components. Specifically, WES fails to analyze untargeted regions, such as introns and regulatory regions of genes. Furthermore, WES cannot identify certain structural variations, particularly those that are exceptionally balanced, such as inversions and translocations. Consequently, the absence of a positive genetic finding does not rule out the possibility of monogenic disease. Some genes have yet to be linked to any specific clinical phenotype. Reanalysis is typically recommended after one year, as it may reveal the discovery of genetic variants in recently identified clinically related genes [13].

WES analysis also plays a crucial role in influencing clinical management decisions, as illustrated in Figure 2. For patients identified through this analysis, various interventions can be implemented, including a multidisciplinary approach, family counseling, screening for manifestations outside the liver, and the introduction of specialized dietary regimens tailored to individual needs. It is of note that clinical management was implemented only for those patients with a confirmed diagnosis and not based on genetic findings alone.

In clinical practice, chronic liver disease is primarily classified into commonly known etiological categories, including toxic exposure, viral infections, cholestatic diseases, autoimmune disorders, metabolic disorders, and certain genetic conditions. However, this approach has a notable drawback in that it fails to account for a broader range of underlying genetic disorders that may be overlooked within these general phenotypes.

An additional critical aspect is the context of living liver donor transplantation (LDLT). In Turkey, the majority of these transplants (80%) are conducted using living liver donors [48,49,50]. Typically, donors are biological relatives. There have been no reports of mortality or morbidity associated with the recurrence of metabolic diseases in heterozygous donors, except in specific cases such as ornithine transcarbamylase deficiency, protein C deficiency, hypercholesterolemia, protoporphyria, and Alagille syndrome [51]. As genetic data continue to expand and enhance our understanding, WES could further identify new pathogenic monogenic or heterozygous carrier mutations, thereby improving outcomes for both recipients and donors following LDLT. This is one of the first studies to apply whole-exome sequencing in a well-characterized cohort of adult patients with unexplained liver disease in Turkey. It demonstrates the diagnostic potential of WES in adult hepatology, revealing clinically relevant genetic variants that can inform patient management and treatment.

However, this study has several important limitations. First, we did not perform functional validation studies or familial segregation analyses. Although variant classification was based on OMIM annotations and genotype–phenotype correlations, the absence of experimental evidence or inheritance data limits the strength of causal interpretations for VUS variants. Future studies should include functional assays and family-based sequencing to improve diagnostic accuracy, especially in cases involving VUS.

Second, the interpretation of VUS is particularly challenging in underrepresented populations, such as our Turkish cohort, due to limited representation in global reference databases. Expanding population-specific genomic resources and incorporating transcriptomic or proteomic analyses may help improve variant interpretation in future research.

Additionally, WES has inherent technical limitations, as it does not capture non-coding regions or detect certain structural variants, such as balanced translocations or deep intronic changes. As a result, some pathogenic variants may have been missed. WGS and complementary techniques, including long-read sequencing, could offer more comprehensive genomic coverage.

Finally, our cohort was heterogeneous, comprising patients with varying liver disease phenotypes. To better understand genetic contributions within specific subtypes, larger-scale studies focusing on more homogeneous groups such as those with hepatic steatosis are needed.

## 5. Conclusions

This exploratory profiling pilot study highlights the potential utility of WES in identifying clinically relevant genetic variants in adults with unexplained liver disease. While not aimed at novel gene discovery, the study underscores the value of WES in uncovering rare known genetic conditions with hepatic involvement, particularly in patients with atypical or syndromic features.

However, the clinical utility of WES relies on confirmatory evidence beyond genetic data. VUS remain challenging, especially in underrepresented populations. Functional validation, family-based studies, and expanded genomic databases are needed to improve interpretation. Future larger, phenotype-focused studies will be essential to determine the diagnostic value and clinical impact of WES in adult hepatology.

## Figures and Tables

**Figure 1 diagnostics-15-02010-f001:**
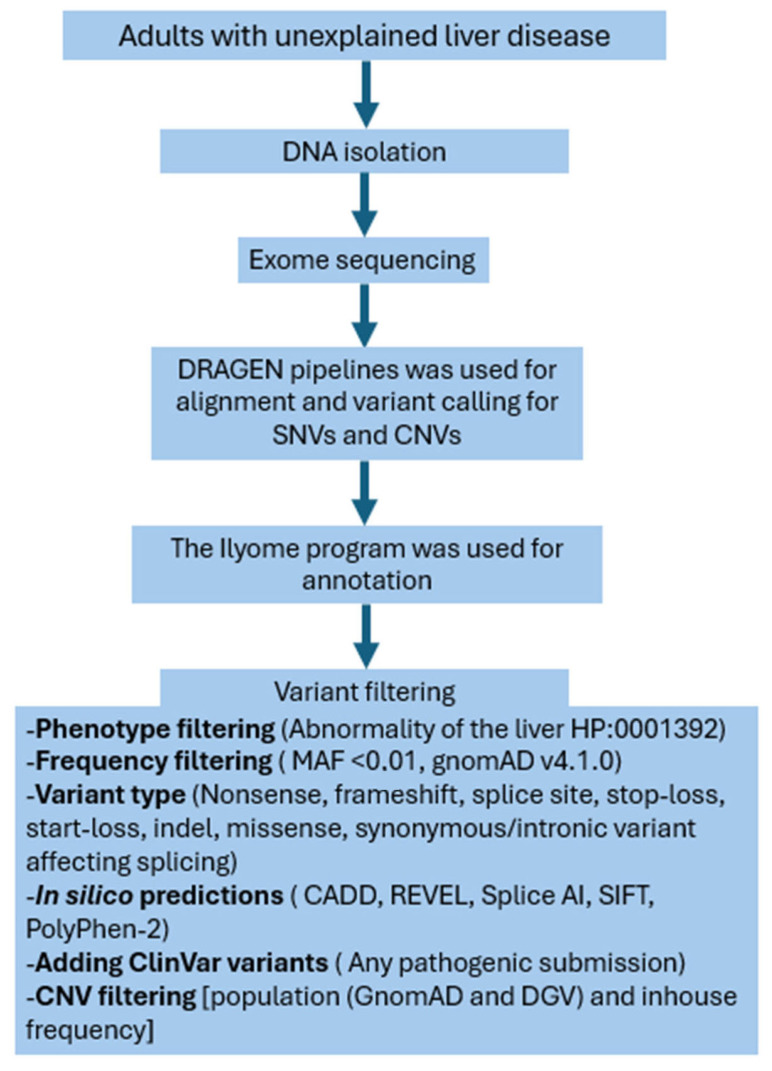
Flowchart of genetic analysis.

**Figure 2 diagnostics-15-02010-f002:**
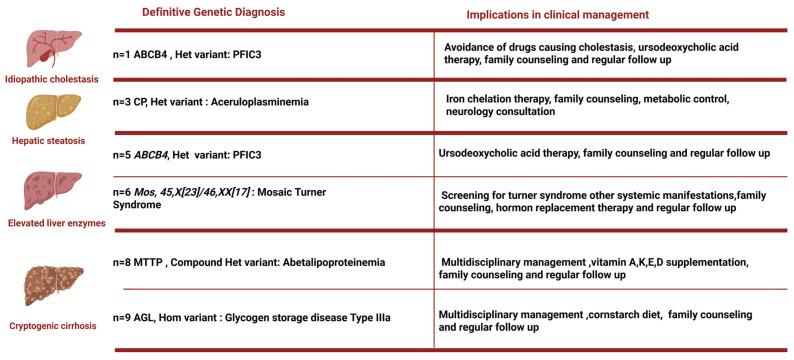
Implications in clinical management after genetic diagnosis for each patient. Het. = heterozygous, Hom. = homozygous.

**Table 1 diagnostics-15-02010-t001:** Demographics and characteristics of total patients (*n* = 53) and patients with definitive diagnosis (*n* = 6).

Variable	Total Patients	Number of Patients with Definitive Genetic Diagnosis	*p*-Value
Female/male, n (%)	18 (34%)/35 (53%)	3 (50%)/3 (50%)	
Age	34 ± 10	31 ± 9	
Family history of liver disease, n (%)	18 (34%)	1 (17%)	*p* = 0.342
Consanguineous marriage, n (%)	23 (43%)	3 (44%)	*p* = 0.729
Cirrhosis, n (%)	7 (13%)	2 (33%)	
Primary liver phenotype, n (%)			
*Idiopathic cholestasis*	6 (11%)	1 (22%)	
*Hepatic steatosis*	28 (53%)	1 (22%)	
*Elevated liver enzymes*	12 (23%)	2 (22%)	
*Cryptogenic cirrhosis*	7 (13%)	2 (34%)	
BMI	24.8	24.5	
Comorbidities			
*Hypothyroidism*	3 (6%)	0 (0%)	
*Hyperlipidemia*	1 (2%)	1 (17%)	
*Type 2 diabetes mellitus*	1 (2%)	1 (17%)	
*Hypertension*	1 (2%)	0 (0%)	
*Chronic kidney disease*	0 (0%)	0 (0%)	
*Chronic lung disease*	0 (0%)	0 (0%)	
Medication history			
*L-thyroxine*	3 (6%)	0 (0%)	
*Metformin*	1 (2%)	1 (2%)	
*Calcium-channel blockers*	1 (2%)	0 (0%)	

BMI = body mass index.

**Table 2 diagnostics-15-02010-t002:** Genetic, clinical, and diagnostic information of the patients with VUS or pathogenic/likely pathogenic variants.

Patient	Clinical Findings	Gene	Transcript	Variant	Frequency (GnomAD v.4.1.0)	Zygosity	Interpretation (ACMG Class)	Genetic Diagnosis	Related Disease Mode of Inheritance (OMIM)
P1	İdiopathic cholestasis	*ABCB4*	NM_000443.4	c.2177C > T p.Pro726Leu	f = 0.00001301	Het	P**	D	Cholestasis, progressive familial intrahepatic 3, AR
P2	İdiopathic cholestasis	*VPS33B*	NM_018668	c.277C > T p.Arg93Ter	f = 0.00001301	Het	P**	P*	Cholestasis, progressive familial intrahepatic, 12, AR
P3	Hepatic steatosis	*CP*	NM_000096.4	c.1948G > A p.Gly650Arg	f = 0.00001301	Het	LP	D	Aceruloplasminemia, AR
		*APOB*	NM_000384.3	c.29C > T p.Ala10Val	f = 0.000002833	Hom	VUS	P*	Hypercholesterolemia, familial, 2, AD
				c.2258G > A p.Gly753Glu	f = 0.0001413	Hom	VUS	P*	
		*KLF11*	NM_003597.5	c.42 + 3G > C	f = 7.255 × 10^−7^	Het	VUS	-	Maturity-onset diabetes of the young, type VII, AD
P4	Hepatic steatosis	*TMEM199*	NM_152464.3	c.532C > T p.Arg178Trp	f = 0.0002225	Hom	VUS	P*	Congenital disorder of glycosylation, type IIp, AR
P5	Elevated liver enzymes	*ABCB4*	NM_000443.4	c.716C > T 5 p.Ser239Leu	f = 0.00000805	Hom	LP	D	Cholestasis, progressive familial intrahepatic 3, AR
P6	Elevated liver enzymes	CNV analysis: chr X deletion, confirmed by karyotype analysis: Mos 45,X[23]/46,XX[17]						D	Mosaic Turner syndrome
		*NBAS*	NM_015909.4	c.5465_5467delinsCTT p.Asn1822_Ile1823de linsThrPhe	f = 0	Het	VUS	-	Infantile liver failure syndrome 2, AR
		*ABCB4*	NM_000443.4	c.217C > G p.Leu73Val	f = 0.001106	Het	VUS	P*	Cholestasis, progressive familial intrahepatic 3, AR
P7	Cryptogenic cirrhosis	*FOCAD*	NM_001375567.1	c.1507C > G p.Pro503Ala	f = 0.00001813	Hom	VUS	P*	Liver disease, severe congenital, AR
P8	Cryptogenic cirrhosis	*MTTP*	NM_001386140.1	c.59_61 + 14del	f = 0	Het	LP	D	Abetalipoproteinemia, AR
				c.1946A > G p.Asn649Ser	f = 0	Het	LP	D	
P9	Cryptogenic cirrhosis	*AGL*	NM_000642.3	c.889A > T p.Lys297	f = 6.204 × 10^−7^	Hom	P**	D	Glycogen storage disease IIIa, AR

D = definitive, P* = possible contribution, P = patient, Het = heterozygous, Hom = homozygous, AR = autosomal recessive, AD = autosomal dominant, P** = pathogenic, LP = likely pathogenic, VUS = Variant of Uncertain Significance, chr: chromosome.

## Data Availability

The datasets generated and analyzed during the current study are available in the Science Data Bank repository: https://www.scidb.cn/en/anonymous/aklmeU1u (10 April 2025.)

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
