# Peer review of "Exome Sequencing in Adults with Unexplained Liver Disease: Diagnostic Yield and Clinical Impact"

_diagnostics, 2025, doi:10.3390/diagnostics15162010_

Round 1

Reviewer 1 Report

Comments and Suggestions for Authors

In this paper, the Authors evaluated the diagnostic yield of exome sequencing in a cohort of adults with unexplained liver disease. The topic is of interest, however there some many issues to address.

Introduction

-the Authors stated that the underlying causes of liver disease remains unknown in 14-30% of patients; please cite references citing this

-please better specify the clinical indications to perform genetic analysis, as described in EASL guideline, to be cited

-please update the references regarding ES in adults with unexplained liver disease: there are other papers beside that cited by the Authors.

Methods

-it is not clear if the Authors performed whole exome sequencing or clinical exome sequencing. It should be useful to report, as supplementary data, the analyzed genes. It is not clear if CNV have been evaluated: please accordingly update the flowchart.

-please better describe the phenotypes evaluated

Results

-the description of genetic results is confusing: it is not clear which patients have a definitive diagnosis, in particular, was P3 considered definitive or possible?

-in Table 2 add the variant ID and the frequency according to gnomAD for each variant; only report the associated disease consistent with the diagnosis (ie PFIC, cholestasis of pregnancy or gallbladder?) with the relative OMIM ID; the “genetic diagnosis” should refer to the patient and not to the single detected variant

-in Table 1 compare the total patients vs those with a definitive diagnosis and not with all patients with genetic findings

-the definition of “possible diagnosis” is debatable; the detected variants are possibly involved in phenotype determination but these are not diagnosis.

-in “study cohort” paragraph the last sentence is not clear

-for P3 discuss the involvement of CP variants in hyerferritinemia and steatosis (Corradini, JHep 2021)

Discussion

Modify accordingly to revised results

Author Response

Comment: The Authors stated that the underlying causes of liver disease remains unknown in 14-30% of patients; please cite references citing this.

Answer: Reference was added (43)

Please better specify the clinical indications to perform genetic analysis, as described in EASL guideline, to be cited

Answer. Has been added to the introduction section with EASL guideline reference (53-56)

Please update the references regarding ES in adults with unexplained liver disease: there are other papers beside that cited by the Authors

Answer: Other studies have been added to the introduction section. (47) (66).

it is not clear if the Authors performed whole exome sequencing or clinical exome sequencing. It should be useful to report, as supplementary data, the analyzed genes. It is not clear if CNV have been evaluated: please accordingly update the flowchart.

Answer: Whole exome sequencing was performed in all patients. It was replaced with the exome sequencing in the text. The analyzed gene list was clarified in the Methods section and was included as a supplementary file number 3 . CNV analysis was performed in all patients. It was explained in the method section and illustrated in the flowchart (added parts are SNV and CNV, phenotype filtering =abnormality of the liver HP:0001392 and CNV filtering population (GnomAD and DGV) and inhouse frequency]). (102-123)

Please better describe the phenotypes evaluated

Phenotypes were more detailly described (137-154)

the description of genetic results is confusing: it is not clear which patients have a definitive diagnosis, in particular, was P3 considered definitive or possible?

The definition of possible diagnosis was changed to possible genetic contribution (164). P3 was considered as definitive diagnosis for hypoceruloplasminemia, while the other genes could be possible genetic contributions.

in Table 2 add the variant ID and the frequency according to gnomAD for each variant; only report the associated disease consistent with the diagnosis (ie PFIC, cholestasis of pregnancy or gallbladder?) with the relative OMIM ID; the “genetic diagnosis” should refer to the patient and not to the single detected variant

Answer: In Table 2, the frequency according to gnomAD for each variant was added, and only associated diseases consistent with the diagnosis were mentioned.

Table 1 compare the total patients vs those with a definitive diagnosis and not with all patients with genetic findings

Answer: The analysis was changed only for defitinive diagnosis and the percentages and p values were changed accordingly. 181-186. Table-1 values were changed.

The definition of “possible diagnosis” is debatable; the detected variants are possibly involved in phenotype determination but these are not diagnosis.

Answer: We changed it to possible genetic contribution and removed the patients from the clinical implications figure, too

In “study cohort” paragraph the last sentence is not clear

Answer : The sentence was changed, and grammar errors and english were improved overall

For P3 discuss the involvement of CP variants in hyerferritinemia and steatosis (Corradini, JHep 2021)

Answer: CP variants involvement in hyperferritinemia and steatosis was addet to the discussion. (250-252)

Discussion

Modify accordingly to revised results

Answer: Modification was done accordingly.

Reviewer 2 Report

Comments and Suggestions for Authors

This is an insightful and potentially impactful study that explores the genetic basis of idiopathic liver disease in adults using exome sequencing (ES). The authors analyzed 53 patients and identified several gene variants that may contribute to previously unexplained liver abnormalities. However, the manuscript requires several critical clarifications:

  1. The manuscript does not specify the location of sample collection or the ethical clearance details. These are essential and should be explicitly stated in both the Abstract and Methods sections.
  2. The Methods mention the use of liver imaging (ultrasound, CT, or MRCP) to exclude known etiologies, but no results from these investigations are reported. Please summarize the key radiological findings for the entire cohort.
  3. It would be highly informative to include a comprehensive table listing all laboratory parameters used to rule out viral, autoimmune, and metabolic liver diseases. This table should present the actual values, relevant cut-offs, and reference ranges where applicable. Currently, the manuscript only notes that tests were performed with negative outcomes, without sufficient detail. 
  4. When reporting the identified gene variants, did the authors compare them against allele frequencies in a healthy Turkish population? What measures were taken to ensure these variants are not common polymorphisms unrelated to liver disease? Clarification is needed, including whether databases such as GnomAD were used as reference datasets.
  5. The flowchart of the study design should be revised to clearly show the number of participants at each step: before and after exclusions, along with detailed inclusion and exclusion criteria.
  6. What clinical and histological criteria were used to define cryptogenic cirrhosis?
  7. How is idiopathic cholestasis defined? Given that cholestasis implies ductal pathology, were MRCP or ERCP performed in these cases to assess biliary tract patency?
  8. The inclusion of patients with hepatic steatosis requires further justification. What specific abnormalities were observed in this subgroup, and were lipid profiles or metabolic evaluations conducted? 
  9. Table 1 should be expanded to include relevant baseline data such as comorbidities, body weight, BMI, and medication history. The current table is insufficient to contextualize the cohort’s clinical background.
  10. For patient P3, who carries a KLF11 variant linked to MODY, please report glucose-related parameters such as fasting blood glucose, HbA1c, and oral glucose tolerance test results. 
  11. All laboratory parameters (e.g., ALT, AST, ALP, GGT) must be reported with proper units throughout the manuscript. For example, ALT-200, AST-110, ALP-156, GGT-220, what are the units of these value?
  12. The term “exome sequencing” should be revised to “whole exome sequencing” (WES) for accuracy and consistency with standard terminology. 
  13. Gene symbols mentioned in the abstract (and main text) should be italicized.
  14. All abbreviations must be defined upon their first use.

In general, the details of the subjects and the examination findings must be included before this work can be evaluated properly. If space constraints are a concern, the authors are encouraged to include supplementary materials.

Author Response

The manuscript does not specify the location of sample collection or the ethical clearance details. These are essential and should be explicitly stated in both the Abstract and Methods sections.

Answer: Ethical statement and location definition was added to the method section.128-134 . This statements weren’t added to the abstract as Diagnostics sees it sufficient to be explained as a footnote.

The Methods mention the use of liver imaging (ultrasound, CT, or MRCP) to exclude known etiologies, but no results from these investigations are reported. Please summarize the key radiological findings for the entire cohort.

Answer: Summary of those findings were added. 96-100

Answer:

It would be highly informative to include a comprehensive table listing all laboratory parameters used to rule out viral, autoimmune, and metabolic liver diseases. This table should present the actual values, relevant cut-offs, and reference ranges where applicable. Currently, the manuscript only notes that tests were performed with negative outcomes, without sufficient detail. 

Answer: Laboratory parameters which were used for all of the 53 patients are added as a supplementary material-1.

When reporting the identified gene variants, did the authors compare them against allele frequencies in a healthy Turkish population? What measures were taken to ensure these variants are not common polymorphisms unrelated to liver disease? Clarification is needed, including whether databases such as GnomAD were used as reference datasets.

Answer: As previously stated in the methods section and flowchart, VAF filtering was performed using the GnomAD database (MAF ≥0.01). The filtering including the Turkish population could not be performed due to insufficient data. However, our in-house database was used, and this was also added to the methods section. (102-123)

The flowchart of the study design should be revised to clearly show the number of participants at each step: before and after exclusions, along with detailed inclusion and exclusion criteria.

Answer : Flowchart of patients design is added as supplementary data 2

What clinical and histological criteria were used to define cryptogenic cirrhosis?

Answer: Cryptogenic cirrhosis was defined as cirrhosis with no identifiable serologic evidence of viral, autoimmune, or metabolic liver disease, no vascular abnormalities on imaging, and nonspecific findings on liver biopsy. Histologically, there were no features suggestive of a specific liver disease, such as interface hepatitis (indicative of autoimmune hepatitis) or cholestatic injury for example (seen in conditions like primary biliary cholangitis or primary sclerosing cholangitis). 133-156

How is idiopathic cholestasis defined? Given that cholestasis implies ductal pathology, were MRCP or ERCP performed in these cases to assess biliary tract patency?

Answer: Definition of cholestasis was added, MRCP was normal in all of those patients, those patients had cholestasis at the microscopic levels in liver biopsy, no gross extrahepatic cholestasis cause (stone, tumor, IgG-4 related disease etc) was present. 133-156

The inclusion of patients with hepatic steatosis requires further justification. What specific abnormalities were observed in this subgroup, and were lipid profiles or metabolic evaluations conducted? 

Answer: The definition of hepatic steatosis was added. We included patients who met the criteria for MASLD (i.e., the presence of at least one cardiometabolic risk factor) along with extrahepatic features suggestive of a monogenic disorder. Among them, only Patient N3 met these criteria and exhibited significant neurological findings. Patients with MASLD, Alcohol-Related Liver Disease (ALD), MetALD, or drug-induced steatotic liver disease were excluded if they lacked prominent extrahepatic manifestations. All patients' metabolic profiles—including HbA1c, LDL, HDL, and triglyceride levels (see Supplementary Material 1)—were reviewed to assess whether they fulfilled the criteria for MASLD or ALD. 133-156

Table 1 should be expanded to include relevant baseline data such as comorbidities, body weight, BMI, and medication history. The current table is insufficient to contextualize the cohort’s clinical background.

Answer: Table-1 was revised according to addded comorbiditites, BMI and medication history. As our patients cohort is young, the amount of comorbidity is less.

For patient P3, who carries a KLF11 variant linked to MODY, please report glucose-related parameters such as fasting blood glucose, HbA1c, and oral glucose tolerance test results. 

Answer: Has been added in the text (254) and in the supplementary data 1.

All laboratory parameters (e.g., ALT, AST, ALP, GGT) must be reported with proper units throughout the manuscript. For example, ALT-200, AST-110, ALP-156, GGT-220, what are the units of these value?

Answer: Units have been added both in text and supplementary data (249, 290)

The term “exome sequencing” should be revised to “whole exome sequencing” (WES) for accuracy and consistency with standard terminology. 

Answer: The term “exome sequencing” was replaced with “whole exome sequencing” (WES) in the text.

Gene symbols mentioned in the abstract (and main text) should be italicized.

Answer: Gene symbols were italicized in the text.

All abbreviations must be defined upon their first use.

Answer : All abbreviations has been defined upon their first use

Round 2

Reviewer 2 Report

Comments and Suggestions for Authors

Thank you for sending the revision and rthe response to my previous comments, some have been resolved but some issues remain:

  1. "This statements weren’t added to the abstract as Diagnostics sees it sufficient to be explained as a footnote.", while this response is acceptable for the ethical clearance, the location and date of data collection must be properly described in the abstract. This is important to identify the population included in this study, including the race and ethnics.
  2. "ALT-136 (<35 U/L) and AST-100 (<35 U/L)" This kind of writing remain unchanged. What are these ALT-136 and AST-100, are they the names of new biomarkers? The authors must have seen other publications while writing this manuscript, has someone wrote this way as well?
  3. "This study aimed to evaluate the diagnostic value of ES in adults with unexplained liver disorder." What is this ES? 
  4. "Although various clinical fields utilize genomic analysis for detecting rare variants", the authors could add this example as well (PMID: 40257486). 
  5. In the introduction, the authors must also highlight the advantages of WES over other means of NGS methods. Why is it used, not WGS, or targeted sequencing for instance? How sensitive it is compared to others?
  6. The authors must add the aims of the study at the end of the introduction. 
  7. "There were no statistically significant differences in family history of liver disease (p = 0.342) or consanguineous marriage (p = 0.729) between the two groups." The authors must show all the p values in Table 1. These values out of sudden appeared in the text but nowhere in the referenced table. 
  8. In Table 2, the authors showed that the found mutations are associated with some diseases and abnormalities. For exampla, P9 is linked to glycogen storage disease. Did they actually confirm that indeed the patients had those diseases? 
  9. Figure 2 is actually insightful if indeed these patients have been confirmed to have these diseases. Otherwise, these diagnosese remain a speculation and therefore, medical interventions should not be taken solely based on the discovered mutations.
  10. The authors must add a new section about the limitations of this study. They should extensively expose their limitations and suggest solutions to these issues.
  11. At this point, I believe that this is a valuable profiling study, but it does not have clinical value yet until those presumed diagnoses based on mutated genes. This must be explicitly declared somewhere in the discussion, limitations and conclusion. 
Comments on the Quality of English Language

Moderate typesetting is needed, some typos are present. The use of MDPI editing service could be needed. 

Not sure if this exists "In instances"

Author Response

Thank you for sending the revision and the response to my previous comments, some have been resolved but some issues remain:

Thank you for your insightful anaylsis and critique.

  1. "This statements weren’t added to the abstract as Diagnostics sees it sufficient to be explained as a footnote.", while this response is acceptable for the ethical clearance, the location and date of data collection must be properly described in the abstract. This is important to identify the population included in this study, including the race and ethnics.

Thank you for your suggestion. Location, date of data collection and ethnicity have been added to the abstract. (25-27)

  1. "ALT-136 (<35 U/L) and AST-100 (<35 U/L)" This kind of writing remain unchanged. What are these ALT-136 and AST-100, are they the names of new biomarkers? The authors must have seen other publications while writing this manuscript, has someone wrote this way as well?

Thank you for your suggestion. We apologize for overlooking  the abbreviation, has been changed accordingly. (264-265)

  1. "This study aimed to evaluate the diagnostic value of ES in adults with unexplained liver disorder." What is this ES? 

Thank you for your critique. It has been changed to WES. (25)

  1. "Although various clinical fields utilize genomic analysis for detecting rare variants", the authors could add this example as well (PMID: 40257486). 

The reference has been added into the article (47)

  1. In the introduction, the authors must also highlight the advantages of WES over other means of NGS methods. Why is it used, not WGS, or targeted sequencing for instance? How sensitive it is compared to others?

Thank you for your suggestion, this has been added to the introduction section as wished (46-55)

  1. The authors must add the aims of the study at the end of the introduction. 

 Thank you for your suggestion, the aims of the study were more clearly described as wished (76-84)

  1. "There were no statistically significant differences in family history of liver disease (p = 0.342) or consanguineous marriage (p = 0.729) between the two groups." The authors must show all the p values in Table 1. These values out of sudden appeared in the text but nowhere in the referenced table. 

 Has been added to the table-1 (213-214)

  1. In Table 2, the authors showed that the found mutations are associated with some diseases and abnormalities. For example, P9 is linked to glycogen storage disease. Did they actually confirm that indeed the patients had those diseases? 

Thank you very much for your thoughtful and constructive comments. You are absolutely right to highlight the importance of a multidisciplinary approach, and we sincerely apologize for the initial omission in our manuscript. All of our patients were evaluated within a multidisciplinary team consisting of pediatric and adult hepatologists, pediatric metabolic specialists, neurologists, pathologists, and clinical geneticists. This has been explicitly added to the revised manuscript. (184-186)

As described in the Clinical Phenotype section (lines 170–179), a definitive diagnosis was assigned only when a pathogenic or likely pathogenic genetic variant consistent with the clinical phenotype was identified. For example, Patient 9 was considered to have glycogen storage disease based on a combination of clinical features—proximal myopathy, hyperuricemia, hypoglycemia—as well as histologic evidence of vacuolar myopathy on muscle biopsy, alongside liver cirrhosis.

Regarding the remaining three patients with strong clinical phenotypes but only variants of uncertain significance (VUS), we refrained from assigning a definitive diagnosis in line with our stringent classification criteria.

Importantly, our rate of definitive diagnoses is notably lower compared to other studies in the literature (e.g. https://pubmed.ncbi.nlm.nih.gov/37566928/ https://pubmed.ncbi.nlm.nih.gov/37566928/). This reflects our deliberately conservative approach, aimed at minimizing overinterpretation of uncertain genetic findings.

We appreciate the opportunity to clarify these points and have revised the manuscript accordingly.

  1. Figure 2 is actually insightful if indeed these patients have been confirmed to have these diseases. Otherwise, these diagnosese remain a speculation and therefore, medical interventions should not be taken solely based on the discovered mutations.

We thank the reviewer for the valuable feedback. To address this concern, we have ensured that Figure 2 includes only those patients whose diagnoses were confirmed through a multidisciplinary review, integrating genetic findings with clinical, biochemical, and histopathological data. Additionally, we have clarified in the main text that any changes in clinical management were implemented only for those patients with a confirmed diagnosis, and not based on genetic findings alone. (432-433)

  1. The authors must add a new section about the limitations of this study. They should extensively expose their limitations and suggest solutions to these issues.

We appreciate the reviewer’s suggestion. We have now added a dedicated section on the limitations of our study. These include the lack of functional validation studies, the absence of segregation analysis in some cases, and the heterogeneity of the cohort in terms of clinical presentation. We have also discussed potential approaches to address these limitations in future studies, such as incorporating functional assays, expanding cohort size, and utilizing family-based sequencing strategies. (457-475)

  1. At this point, I believe that this is a valuable profiling study, but it does not have clinical value yet until those presumed diagnoses based on mutated genes. This must be explicitly declared somewhere in the discussion, limitations and conclusion. 

We thank the reviewer for their constructive assessment. We fully agree that the clinical utility of our findings is dependent on confirmatory evidence beyond genetic data alone. As such, we have revised the discussion, limitations, and conclusion sections to explicitly state that this is an exploratory, pilot profiling study designed to demonstrate the potential utility of WES in clinical hepatology. Our aim was not to discover novel disease genes but rather to highlight known rare genetic conditions that may present with hepatic involvement.  (481-490)

In instances have been changed to “In cases”, english grammar and sentences has been rechecked and improved.

Thank you very much for your critique and contribution.